# Research on Anal Squamous Cell Carcinoma: Systemic Therapy Strategies for Anal Cancer

**DOI:** 10.3390/cancers13092180

**Published:** 2021-05-01

**Authors:** Ryan M. Carr, Zhaohui Jin, Joleen Hubbard

**Affiliations:** Department of Medical Oncology, Mayo Clinic, Rochester, MN 55903, USA; carr.ryan@mayo.edu

**Keywords:** anal squamous cell carcinoma, human papillomavirus, chemoradiotherapy, immunotherapy, papillomavirus vaccines, PI3K, mTOR

## Abstract

**Simple Summary:**

Anal cancer is rare with an estimated 9000 new cases predicted to occur in the United States in 2021. However, rates of new anal cancer cases and deaths from the disease are increasing by about 2% and 3% per year respectively. In light of these trends it is critical to better understand the nature of this disease and progress in its management. The present review focuses on the history and development of the role of systemic therapy in the treatment of anal cancer. Major trials establishing the role of chemotherapy in the management of locoregional and metastatic anal cancer are summarized. In addition, the rapidly evolving role of immunotherapy is discussed. Finally, major insights into the molecular pathobiology of anal cancer and opportunities for advancement in precision medicine in treatment of the disease.

**Abstract:**

Anal squamous cell carcinoma (ASCC) is a rare malignancy, with most cases associated with human papilloma virus and an increased incidence in immunocompromised patients. Progress in management of ASCC has been limited not only due to its rarity, but also the associated lack of research funding and social stigma. Historically, standard of care for invasive ASCC has been highly morbid surgical resection, requiring a permanent colostomy. Surgery was associated with disease recurrence in approximately half of the patients. However, the use of chemotherapy (5-fluorouracil and mitomycin C) concomitantly with radiation in the 1970s resulted in disease regression, curing a subset of patients and sparing them from morbid surgery. Validation of the use of systemic therapy in prospective trials was not achieved until approximately 20 years later. In this review, advancements and shortcomings in the use of systemic therapy in the management of ASCC will be discussed. Not only will standard-of-care systemic therapies for locoregional and metastatic disease be reviewed, but the evolving role of novel treatment strategies such as immune checkpoint inhibitors, HPV-based vaccines, and molecularly targeted therapies will also be covered. While advances in ASCC treatment have remained largely incremental, with increased biological insight, an increasing number of promising systemic treatment modalities are being explored.

## 1. Introduction

Squamous cell carcinoma is the most common malignant histologic subtype affecting the anal canal (ASCC). The anal canal anatomically occurs between the anorectal junction proximally and the anal verge distally and is approximately 3–5 cm in length. The transitional zone between the columnar epithelium of the rectum and the unkeratinized squamous cells of the anal mucosa, and proximal to the dentate line, is the site of most cases of ASCC [1]. Etiologically, the vast majority of cases are associated with human papilloma virus (HPV) infection and its incidence is significantly elevated in immunocompromised patients [2,3]. There were about 8300 new cases of cancer involving the anus in the United States in 2019. While ASCC is rare, making up only 2.5% of gastrointestinal malignancies, its incidence continues to increase [4]. Social stigma, rarity of the disease and associated lack of research funding have contributed to under-recognition of the malignancy and hampered progress in its management. In this review, the history and role of systemic therapy in the management of ASCC will be discussed. In the age of precision medicine, the evolving role of targeted agents, immunotherapy, and other novel treatment modalities will be explored.

## 2. Chemotherapy for Locoregional Disease

Historically, standard of care for invasive ASCC was abdominal perineal resection (APR). Given it involves removal of the anorectum, APR requires a permanent colostomy. Even accepting such morbidity, five-year survival after APR only ranges between 40% and 70% [5,6]. However, in 1974, three case reports published by Nigro et al. proved influential in making chemoradiation standard of care. Two patients received chemoradiation with 5-fluorouracil (5-FU) and mitomycin C (or poriferomycin), while one received radiation alone. Tumor regression was seen in all three cases with no evidence of residual disease appreciated on subsequent surgical resection. One patient refused APR and reportedly remained disease-free [7]. Such observations suggested chemoradiation could potentially obviate surgical resection and its associated morbidity. This was validated in later prospective studies [8,9,10]. A summary of landmark study results in the management of locoregional ASCC is provided in Table 1. An EORTC trial in 1997 was one of the first randomized phase III trials investigating 5-FU and mitomycin with concomitant radiation for a five-week treatment course vs. radiation alone in patients with locally advanced anal cancer. The trial enrolled 110 patients randomized between the two arms. Results confirmed the role of multimodality treatment with chemoradiation in conferring significantly increased complete response (CR) rates, lower locoregional recurrence rates, higher locoregional control, and longer colostomy-free interval [9]. Similarly, the larger ACT I phase III study also compared radiation or chemoradiation arms. This confirmed the superiority of chemoradiation as it conferred reduced local failure rate [10], while median overall survival (OS) differences could not be discerned until long-term follow-up published in 2010. This revealed reduced locoregional relapse and ASCC death with improved OS [11]. Finally, the importance of mitomycin in the chemoradiation regimen was assessed in an intergroup phase III study. Relative to 5-FU alone, the addition of mitomycin improved colostomy-free survival and disease-free survival (DFS) [12]. Taken together, since the 1970s, chemoradiation has remained the standard-of-care for all nonmetastatic ASCC cases given its improved outcomes and reduced morbidity with APR reserved as a salvage therapy.

Attempts to improve on this treatment paradigm have been limited. In a retrospective cohort study including 299 elderly patients (median age of 72) with stage I ASCC, 200 were treated with chemoradiation vs. 99 treated with radiation alone. After propensity-score adjustments, the addition of chemotherapy did not significantly improve OS, DFS, colostomy-free survival or cause-specific survival in this select group [13]. This finding potentially supports de-escalation of therapy in carefully selected patients.

Alternatives to 5-FU and mitomycin have also been explored. For example, the oral fluoropyrimidine prodrug capecitabine has proven to be interchangeable with infusional 5-FU in the treatment of other malignancies such as with gastric cancer in the REAL-2 phase III study [14] or colorectal adenocarcinoma in the X-ACT phase III trial [15]. Several retrospective studies have demonstrated safety and efficacy of capecitabine and mitomycin in locoregional ASCC [16,17,18]. One study included 105 patients with ASCC with 47 treated with 5-FU-based chemoradiation while 58 were treated with capecitabine-based therapy. This demonstrated nonsignificant differences in CR rates, three-year locoregional control, three-year OS and colostomy-free survival [17]. While randomized prospective comparisons are lacking in ASCC, these retrospective findings are comparable to clinical outcomes and safety data from two studies. The EXTRA phase II trial included 31 patients with ASCC receiving chemoradiation with capecitabine and mitomycin and demonstrated a CR rate of 77% [19]. A later phase II, single-arm trial similarly used capecitabine-based chemoradiation in 43 patients with ASCC, demonstrating an 86% locoregional control rate at 6 months [20]. Therefore, capecitabine is considered as an appropriate alternative to infusional 5-FU for locoregional ASCC.

Improving chemoradiation by replacing mitomycin with cisplatin has also been tested. The ACT II trial was a randomized, phase III, open-label study consisting of 940 patients comparing radiation with 5-FU and mitomycin vs. 5-FU and cisplatin. It should be noted that, instead of giving mitomycin at 10 mg/m^2^ for two doses, it was administered at 12 mg/m^2^ as a single dose. There were no significant differences in CR rates nor grade 3–4 adverse effects between the chemotherapy regimens [21]. Therefore, feasibility of treatment escalation was tested in a phase II pilot study in which 19 patients were treated with radiation concomitantly with 5-FU, mitomycin and cisplatin. Unfortunately, given the very high toxicity rates with this regimen, triplet therapy was not considered reasonable [22]. Thus, while chemoradiation with 5-FU and cisplatin is considered an alternative to 5-FU and mitomycin, triplet therapy is deemed too toxic.

## 3. Role of Induction or Maintenance Chemotherapy

Chemoradiation has largely been the standard of care for locoregional ASCC since the 1970s. While the previously highlighted trials firmly support the use of chemoradiation, there have been a few attempts to advance clinical outcomes through the modification of available regimens. Two examples include the addition of either induction or maintenance chemotherapy to chemoradiation. The aforementioned ACT II study had a 2 × 2 factorial design assessing the utility of maintenance chemotherapy following chemoradiation. In the two treatment arms, including maintenance chemotherapy, patients received an additional two cycles of fluorouracil with cisplatin at weeks 11 and 14. Of the patients who received cisplatin- and mitomycin-based chemoradiation, 222 and 226 patients, respectively, were randomized to receive maintenance chemotherapy. However, this did not significantly improve three-year PFS [21].

Akin to ACT II, the intergroup RTOG 98–11 study was a phase III trial randomizing 325 patients to chemoradiation with 5-FU and mitomycin and 324 patients to the 5-FU and cisplatin arm. Interestingly, the mitomycin arm resulted in improved five-year DFS and OS [23]. However, interpretation of these results must be made cautiously given patients in the cisplatin arm received induction 5-FU and cisplatin prior to chemoradiation while the mitomycin arm did not. Thus, it is difficult to attribute differences in outcomes purely to comparisons between mitomycin and cisplatin. In fact, in light of the ACT II trial, these results may suggest a detrimental effect of induction chemotherapy. 

The ACCORD 03 study was a phase III trial that directly tested treatment intensification by adding two cycles of induction chemotherapy with 5-FU and cisplatin prior to chemoradiation. The addition of induction chemotherapy caused no significant differences in colostomy-free survival [25]. These studies, in addition to a systematic review, demonstrate no benefit of induction chemotherapy in ASCC management [24]. Taken together, there is no clear role for induction or maintenance chemotherapy in the management of nonmetastatic ASCC.

## 4. Systemic Therapy for Metastatic Disease

Management of locoregional ASCC is largely one-size-fits-all irrespective of precise staging due to the relative rarity of the disease. However, approximately 10–20% of patients treated with curative intent will develop metastatic disease. In addition, less than 10% of patients with ASCC present with de novo metastatic disease [10,26]. Prognosis for these patients is poor with an approximately 30% five-year survival rate [27].

Chemotherapy is routinely offered to patients with metastatic ASCC. In this setting, guidelines have historically recommended a platinum doublet including a fluoropyrimidine as first-line treatment [28,29]. There are limited data supporting the use of leucovorin, fluorouracil and oxaliplatin (FOLFOX) as well as FOLFCIS, effectively a FOLFOX schedule with cisplatin replacing oxaliplatin [30,31]. Nonetheless, until 2018, treatment recommendations have been based upon similar case series and retrospective studies. Table 2 summarizes key prospective trials in the management of ASCC. The Epitopes-HPV02 trial was a single-arm phase II study with nonoperable or metastatic ASCC treated with either standard or modified docetaxel, cisplatin and fluorouracil (DCF and mDCF, respectively). DCF treatment consisted of six cycles of docetaxel (75 mg/m^2^ on day one), cisplatin (75 mg/m^2^ on day one), and fluorouracil (750 mg/m^2^ per day for five days) every three weeks. The mDCF regimen consisted of eight cycles of docetaxel (40 mg/m^2^ on day one), cisplatin (40 mg/m^2^ on day one), and fluorouracil (1200 mg/m^2^ per day for 2 days) every two weeks. Choice of the two treatments was not randomized. Instead, it was determined by the patient’s age and performance status. PFS between the two treatment regimens was not significantly different. However, there were significantly more grade 4 adverse events in those who received DCF vs. mDCF, making the latter a potential first line option for metastatic ASCC [32].

The phase II InterAAct study was the first randomized trial for patients with unresectable, metastatic ASCC. Patients were treated with either carboplatin and paclitaxel or cisplatin. While ORR values between the regimens were comparable, carboplatin and paclitaxel conferred superior median PFS and OS. Furthermore, there was a significant increase in more serious adverse events in the cisplatin and fluorouracil arm [33]. Taken together, while mDCF is a promising option, the higher quality data supports using carboplatin and paclitaxel in the first line for metastatic ASCC.

## 5. The Evolving Role of Immune Checkpoint Inhibitors

The use of immunotherapy has also impacted management of metastatic ASCC. While pembrolizumab is FDA approved for treatment of microsatellite instable or mismatch repair deficient (MSI/dMMR) malignancies, ASCC is rarely MSI/dMMR. However, these tumors tend to have high expression of PD-L1 and/or a high tumor mutation load despite being microsatellite stable, potentially suggesting sensitivity to immune checkpoint inhibitors [36]. Furthermore, the vast majority of ASCC cases (>95%) are associated with HPV infection [37]. Other HPV-related squamous cell carcinomas, such as those afflicting the head and neck, have demonstrated significant response rates to immunotherapy. For example, the phase Ib KEYNOTE-012 study, evaluating safety and efficacy of pembrolizumab in recurrent or metastatic head and neck squamous cell carcinoma, demonstrated an ORR of 18%. Interestingly, the duration of these responses was prolonged, with 85% lasting greater than six months [38]. Collectively, these findings made immune checkpoint inhibitors a promising treatment option for ASCC. 

Thus, the phase Ib KEYNOTE-028 trial was a single-armed study treating 25 patients with anal cancer (24 of which were squamous cell histology) with PD-L1 expression ≥1% by immunohistochemistry with pembrolizumab. The ORR was 17% with a median PFS and OS of 3.0 and 9.3 months, respectively [34]. Nivolumab was used in the NCI9673 single-arm, phase II study in previously treated, metastatic ASCC. This conferred a response rate of 24% [35]. Currently, the NCI9673 multi-institutional phase II study is randomizing patients with previously treated metastatic ASCC in a 1:1 fashion to either nivolumab or nivolumab and ipilimumab (NCT02314169). Thus, as of the writing of this review, guidelines recommend reserving immunotherapy for the second line for metastatic ASCC.

Given the promising findings of these studies, there are several ongoing investigations assessing the earlier use of immune checkpoint inhibitors in ASCC management. Expanding on the aforementioned InterAAct trial, there are two ongoing studies assessing use of immune checkpoint inhibitors in the frontline setting for metastatic disease. EA2176 is a 2:1 randomized phase III trial of carboplatin and paclitaxel with or without nivolumab (NCT02178241). In addition, POD1UM 303/InterAAct 2 is a 1:1 randomized phase III trial of carboplatin and paclitaxel with or without anti-PD-1 antibody, retifanlimab, for newly diagnosed metastatic ASCC (NCT04472429). Finally, EA2165 is a phase II study randomizing patients with high risk ASCC to adjuvant nivolumab vs. observation after completion of definitive chemoradiation with a primary endpoint of DFS. High risk ASCC is considered stage IIB (T3N0M0 only) and stage III (T4 tumors or node positive disease without metastases) disease (NCT03233711). Therefore, immunotherapy continues to significantly alter the landscape of management options for this orphan disease.

## 6. Role of Human Papillomavirus in Treatment Strategies

Given the significant association with HPV infection and ASCC risk, this key feature of the disease’s pathobiology may potentially inform novel therapeutic intervention. HPV type 16 (HPV16) is the genotype most identified making up approximately 81% of cases followed by HPV33 (5.1%), HPV18 (2.2%), and HPV58 (0.7%) [2]. Infection results in expression of oncogenes E6 and E7, which facilitate tumor suppressor dysregulation with degradation of p53 and inhibition of Rb, respectively. Inhibition of Rb results in increased cell proliferation and compensatory upregulation of tumor suppressor p16, a commonly used surrogate marker of HPV positivity [39]. Tumor infiltrating lymphocyte-based therapy has been developed capitalizing on this biological feature of ASCC (and other squamous cell carcinomas). A recent phase I/II study investigated the safety and efficacy of autologous genetically engineered T cells expressing a T-cell receptor directed against HPV16 E6 after conditioning aldesleukin. This was used in patients with previously treated metastatic HPV16-positive malignancies. Of the 12 patients enrolled, four had ASCC. While two of these patients had progressive disease, two had a partial response with no dose-limiting toxicities. Of note, one of the anal cancer cases that developed progressive disease had heterozygous loss of HLA-A*02:01, which is a necessary restriction element for the engineered T cells [40].

Other clinical trials are currently underway, aiming to capitalize on the significant association between HPV infection and ASCC. Not all individuals exposed to HPV ultimately develop malignancies, but through the use of genome wide association study data, transforming growth factor beta (TGFβ) signaling was identified as a pathway significantly associated with cancer formation after infection [41]. Thus, two National Cancer Institute trials are being conducted using M7824 in HPV-associated malignancies including ASCC. M7824 is a fusion protein consisting of a monoclonal antibody against PD-L1 linked to the extracellular domain of the human TGFβ receptor 2, serving as a TGFβ trap. Thus, this would effectively neutralize both PD-1/PD-L1-mediated immunosuppression as well as TGFβ signaling. While one study is investigating its safety and efficacy alone (NCT03427411), another is treating with an HPV vaccine with or without M7824 (NCT04432597). In addition to safety and preliminary efficacy data, the trials also aim to investigate the extent to which CD3+ tumor infiltrating T cells increase with these interventions, a correlative finding hypothesized to serve as a proxy for effectiveness. Finally, an MD Anderson phase II (NCT03439085) study is also treating patients with metastatic HPV-associated malignancies with a therapeutic HPV vaccine (INO-3112) consisting of a DNA plasmid encoding interleukin-12, meant to serve as a potent immunopotentiator of T-cell function. This is being combined with durvalumab, an immune checkpoint inhibitor targeting PD-1. The strong association of ASCC with HPV infection opens opportunities to take advantage of this feature of the disease’s pathobiology for the purposes of novel therapeutic interventions.

## 7. Precision Medicine and Targeted Therapy

### 7.1. Epidermal Growth Factor Receptor Blockade

In the era of cutting-edge bioinformatics and precision medicine, great strides have been made in management of various cancers with the development of molecularly targeted therapeutics. Unfortunately, such advances have eluded treatment of ASCC to date. The most extensively studied targeted treatment modality in ASCC is the blockade of epidermal growth factor receptor (EGFR), which is a type 1 tyrosine kinase transmembrane receptor that can facilitate downstream growth signaling. Interest in EGFR as a potential treatment target stemmed from observed high surface expression in more than 90% of studied ASCC patient biopsies with increased expression associated with tumor progression [42,43]. In another study, approximately 34% of ASCC samples demonstrated elevated EGFR copy numbers due to either amplification or polysomy [44]. Preclinical studies showed promise with murine xenograft models treated with chimeric anti-EGFR monoclonal antibody, cetuximab, demonstrating attenuation of tumor growth [45].

However, EGFR abrogation fell short in clinical trials primarily due to an unacceptable adverse effect profile of cetuximab. Results of these studies are summarized in Table 3. An initial phase I study was conducted incorporating cetuximab into the chemoradiation regimen with 5-FU and cisplatin demonstrated a response rate as high as 95%. However, the study had to be closed prematurely due to high rates of grade 3 and 4 adverse events including radiation dermatitis, diarrhea, venous thromboembolism and infection. In a safety expansion cohort, there were three grade four venous thromboembolism events in rapid succession resulting in study closure [46]. Similarly, the ACCORD 16 also investigated cisplatin, 5-FU and cetuximab. While results were, once again, promising with a one-year PFS of 62% and OS of 92%, this study was also prematurely closed due to 15 serious adverse effects in 14 of the 16 patients on the study [47]. Later, two additional phase II studies incorporating cetuximab to chemoradiation with cisplatin and 5-FU in immunocompetent [48] and HIV-associated ASCC [49] demonstrated high rates (26–32%) grade 4 adverse events. These studies also had 5% and 4% treatment-associated deaths, respectively. Ultimately, addition of cetuximab to definitive chemoradiation was deemed unacceptably toxic. In light of this shortcoming, a more comprehensive molecular characterization of ASCC is necessary to inform alternative targeted therapeutic strategies.

### 7.2. PI3K/Akt/mTOR Signaling Axis

Several efforts to characterize somatic abnormalities in ASCC have been made. Many of these studies include whole exome sequencing (WES), next generation sequencing (NGS) and copy number alteration (CNA) analysis of patient-derived samples of ASCC. Across these investigations, somatic variants in the PI3K/AKT/mTOR (phosphatidylinositol 4,5-bisphosphate 3-kinase/protein kinase B/mammalian target of rapamycin) signaling axis were recurrently identified. PI3K is a plasma membrane-associated lipid kinase consisting of three subunits (alpha, beta and delta) encoded by different genes. Rates of pathogenic mutations identified in the alpha catalytic subunit of PI3K, *PIK3CA*, range between 16% and 40% [37,44,45,50,51,52,53]. Those studies involving genomic hybridization or other CNA analyses demonstrated approximately 63% of ASCC cases with recurrent amplifications or homozygous deletions involving the PI3K/AKT/mTOR pathway [52]. In fact, the most frequent minimal region of gain, occurring in 57% of cases, encompassed the 3q26.32 locus, which contains *PIK3CA* [54,55,56]. In addition, deleterious alterations in the PI3K negative regulator, *PTEN* (phosphatase and tensin homolog), are found in approximately 15% of cases [52,56]. Taken together, these data indicate that PI3K/AKT/mTOR signaling may serve as an ideal therapeutic target in ASCC (Figure 1).

Preclinical models support the important role of the PI3K/AKT/mTOR axis in the pathogenesis of ASCC. First, Stelzer and colleagues used two different mouse models. One was a genetically engineered mouse model (GEMM) expressing HPV oncogenes (E6 and E7) in stratified squamous epithelia. Topical application of carcinogen, dimethylbenz[a]anthracene (DMBA), resulted in ASCC formation in the transgenic mice, whereas none developed in the wildtype mice. In addition, they developed patient-derived murine xenografts. Both models were treated with mTOR inhibitor, rapamycin. Inhibition of mTOR signaling successfully attenuated tumor growth in both models and prophylactic rapamycin reduced the incidence of ASCC formation in the transgenic mouse [57]. Interestingly, rapamycin treatment was noted to result in compensatory MAPK signaling activity potentially identifying an escape mechanism for ASCC. Sun and colleagues developed a GEMM that would spontaneously develop ASCC using an inducible K14-Cre to delete both *Tgfbr1* (TGFβ receptor 1) and *Pten* in squamous epithelia. Similarly, treatment with rapamycin resulted in delayed onset of ASCC in the preventative setting and reduced tumor burden when treating established disease [58]. More recently, different GEMM have been developed to investigate the relative roles of *PIK3CA* activating mutations and HPV oncogenes in ASCC development. This was carried out using mice harboring *Pik3ca* activation mutations (H1047R or E545K) in anal epithelium with or without transgenic E6 and E7 expression followed by DMBA application. While the combined E6, E7 and *Pik3ca* mutations resulted in spontaneous ASCC formation in the absence of DMBA, the presence of *Pik3ca* mutations alone were sufficient to lead to tumor formation with DMBA application. This resulted in treatment with TAK-228 (an mTOR1/2 inhibitor) in patient-derived ASCC organoids and xenografts resulting in reduced tumor size [3]. Interestingly, use of PI3K inhibitor, BYL719, in treatment of xenografts, with targeted sequencing confirming a *PIK3CA* E545K mutation, failed to decrease tumor growth [45]. While no mechanistic investigations were performed to explain this finding, this may be a consequence of the activity of the inhibitor. A basket trial had <20% of *PIK3CA* mutated cases demonstrating a treatment response using the BYL719 inhibitor [59]. These preclinical studies demonstrate the potential promise of PI3K/AKT/mTOR signaling inhibition in the management of ASCC though, given the variety of PI3K and mTOR inhibitors under development, more mechanistic investigations are necessary to clarify optimal treatment strategies.

The likely most effective way to use small molecule inhibitors of PI3K/AKT/mTOR signaling would be in the maintenance setting after definitive chemoradiation or, perhaps, after salvage APR. This is partly due to the relatively cytostatic nature of both PI3K and mTOR inhibitors as demonstrated in the aforementioned preclinical trials. Furthermore, while frequency of *PIK3CA* mutations are relatively unchanged pre- and postchemoradiation, its presence at recurrence after salvage APR confers a poor prognosis relative to wildtype cases [51,60]. Thus, PI3K/mTOR inhibitors may be used to prevent such recurrences. In addition, patients presenting with *TP53* mutations tend to be HPV negative and have worse prognoses [52,61,62,63]. Mutations in *TP53* are also more frequent in recurrent disease [51]. Interestingly, part of the tumor suppressor function of p53 involves suppression of mTOR signaling. A recent study demonstrated that unopposed mTOR signaling may be critical in tumor formation in the context of mutant p53. Treatment with an mTOR inhibitor suppressed tumor formation in *Tp53* null mice [64]. This provides additional rationale for the use of PI3K/mTOR inhibitors in the management of ASCC.

### 7.3. Other Opportunities for Targeted Therapy

While not extensively tested in preclinical or clinical investigations in the context of ASCC to date, genomic profiling reveals additional potential targets for novel treatment strategies. Two pathways recurrently altered are RAS signaling and DNA repair. The former involves activating mutations of *KRAS* (4.3%) and *NRAS* (1.4%) or deactivating mutations in *NF-1* (4.3%) [52]. Despite RAS signaling being commonly aberrantly activated across cancer subtypes, it has been notoriously challenging to “drug”. However, the recent development of small molecule inhibitors such as sotorasib, which successfully inhibit *KRAS^G12C^* driven disease, may represent a breakthrough in abrogation of the signaling axis [65]. Unfortunately, the frequency of this specific somatic mutation in ASCC is not reported and likely low [66]. Thus, efforts to block RAS signaling in ASCC would need to focus on downstream effectors.

In addition, mutations involving DNA repair are found in approximately 10% of ASCC cases. The most common variants identified are in *ATM* (5.7%), *BRCA2* (2.9%) and *BRCA1* (1.4%), raising the possible therapeutic efficacy poly(ADP-ribose) polymerase (PARP) inhibitors [52]. Most investigations into the use of PARP inhibitors have been in squamous cell carcinoma of the head and neck though these are agnostic of tumor genetics. Instead, PARP inhibitor use has been studied as treatments that may potentially synergize with or sensitize one to platinum-based chemotherapy and irradiation, respectively. Interaction with these standard modalities has been demonstrated in multiple experimental models [67,68,69,70]. The Alliance A091101 trial is investigating the addition of the PARP inhibitor, veliparib, to induce carboplatin and paclitaxel in locally advanced head and neck squamous cell carcinoma. While the phase I results have been reported [71], demonstrating tolerability of the treatment regimen, the phase II study is ongoing. It is unclear whether this treatment strategy could be extrapolated to ASCC standard-of-care or whether the 10% of cases harboring DNA damage repair mutations are more likely to respond to PARP inhibitors.

Other recurrent mutations have relatively low frequencies, though still offer potential therapeutic strategies. Various growth factor receptors harbor mutations, resulting in upregulation of downstream signaling including *FGFR2* (4.3%), *FGFR1* (2.9%), *ERBB2* (2.9%) and EGFR (1.4%) [52]. Finally, mutations involving epigenetic regulators have been identified in *SMARCB1* (2.9%) and *SMARCA4* (1.4%). These two genes encode proteins part of a multi-sub-unit chromatin remodeling complex called SWI/SNF (SWItch/Sucrose Nonfermentable) which facilitates histone acetyltransferase, resulting in transcriptional activation. *SMARCA4* encodes the catalytic domain, whereas *SMARCB1* encodes a core subunit of the complex [72]. While the SWI/SNF complex facilitates chromatin accessibility, it is opposed by polycomb repressive complex 2 (PRC2). *EZH2* encodes the catalytic subunit Enhancer of zeste homoglog 2, which confers histone H3 lysine 27 trimethylation (H3K27me3), resulting in gene silencing [73]. With inactivating mutations in ASCC in critical SWI/SNF subunits, unopposed PRC2 activity becomes a potential treatment strategy using EZH2 inhibitors. These ASCC precision medicine opportunities are graphically depicted in Figure 2.

## 8. Conclusions

ASCC is a rare, HPV-associated malignancy increasing in incidence. The rarity of the disease has historically made treatment recommendations based upon prospective, randomized data elusive. Chemoradiation supplanted upfront APR in the 1970s but superior morbidity and mortality outcomes were not validated in prospective trials until approximately twenty years later. To date, 5-FU and mitomycin have remained the chemotherapy of choice. Otherwise, advances in treatment of locoregional disease have eluded investigators with APR continuing to be reserved as a salvage therapy.

The most rapidly evolving treatment paradigm is in the metastatic setting considering use of immune checkpoint inhibitors. While carboplatin and paclitaxel were recently established as the preferred front-line treatment for metastatic disease with the InterAAct trial, other combinations have reasonable activity including cisplatin and 5-FU or mDCF. Immunotherapy is currently reserved for the second-line setting but is being tested in combination with front-line chemotherapy and even in the adjuvant setting following definitive chemoradiation in ongoing trials. The current treatment paradigm for ASCC is summarized in Figure 3.

If successful, immunotherapy could even show promise when combined with definitive chemoradiation. While the efficacy of immune checkpoint inhibitors and other novel immunotherapies in ASCC is cause for hope in advancement of management, targeted therapeutics remain largely unexplored. Targeting EGFR with cetuximab has been abandoned as part of definitive chemoradiotherapy due to severe adverse effects. However, our current understanding of the pathobiology of ASCC has yet to be fully leveraged in the clinical setting. Namely, the preponderance of aberrations in PI3K/AKT/mTOR signaling in ASCC suggests potential new treatment opportunities, especially with several PI3K/mTOR small molecular inhibitors actively being used in practice today. While treatment options for this rare cancer have remained relatively limited for decades, ongoing advancement in understanding of immunology and molecular biology of ASCC continue to open new opportunities to treat those afflicted with this disease.

## Figures and Tables

**Figure 1 cancers-13-02180-f001:**
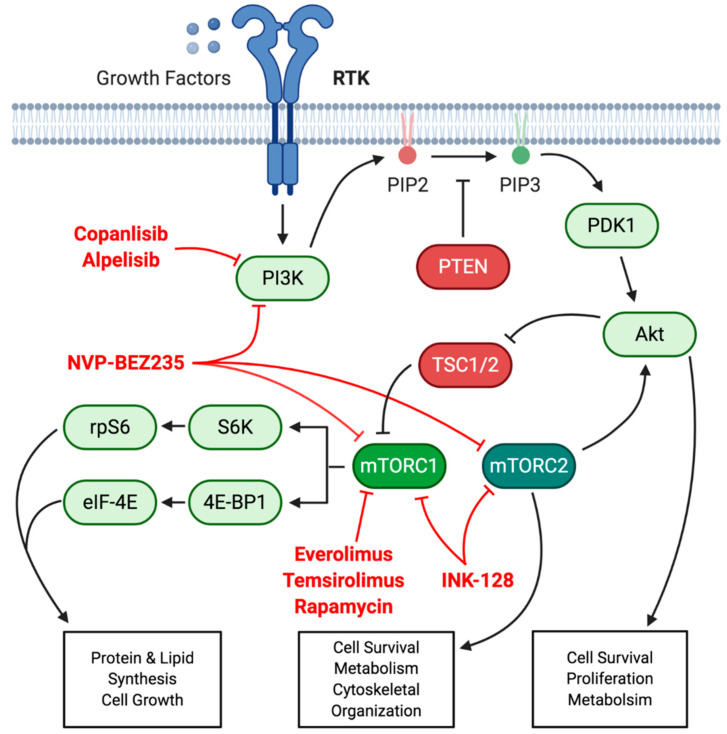
PI3K/Akt/mTOR signaling axis. Growth factors bind to cognate RTK (receptor tyrosine kinase), resulting in downstream phosphorylation and activation of PI3K (phosphatidylinositol 4,5-bisphosphate 3-kinase). PI3K phosphorylates PIP2 (phosphatidylinositol 4,5-bisphosphate) to PIP3 (phosphatidylinositol 1,4,5-triphosphate). Conversion from PIP3 back to PIP2 is catalyzed by tumor suppressor PTEN (phosphatase and tensin homolog). Increased PIP3 facilitates activation of PDK1 (phosphoinositide-dependent kinase-1) and Akt to promote cell survival and proliferation. Part of Akt activity is through downstream activation of mTOR (mammalian target of rapamycin), exists in the context of two complexes (mTORC1 and mTORC2) with different binding partners. mTORC1, through the activation of downstream effectors, facilitates cell growth through promoting protein and lipid synthesis. The activity of mTORC1 is derepressed through inactivation of TSC1/2 by Akt. Indicated in red are several pharmacologic inhibitors of this signaling axis either in clinical practice or currently under investigation. S6K, S6 serine/threonine kinase; rpS6, ribosomal protein S6; 4E-BP1, 4E-binding protein 1; eIF-4E, eukaryotic translation initiation factor 4E.

**Figure 2 cancers-13-02180-f002:**
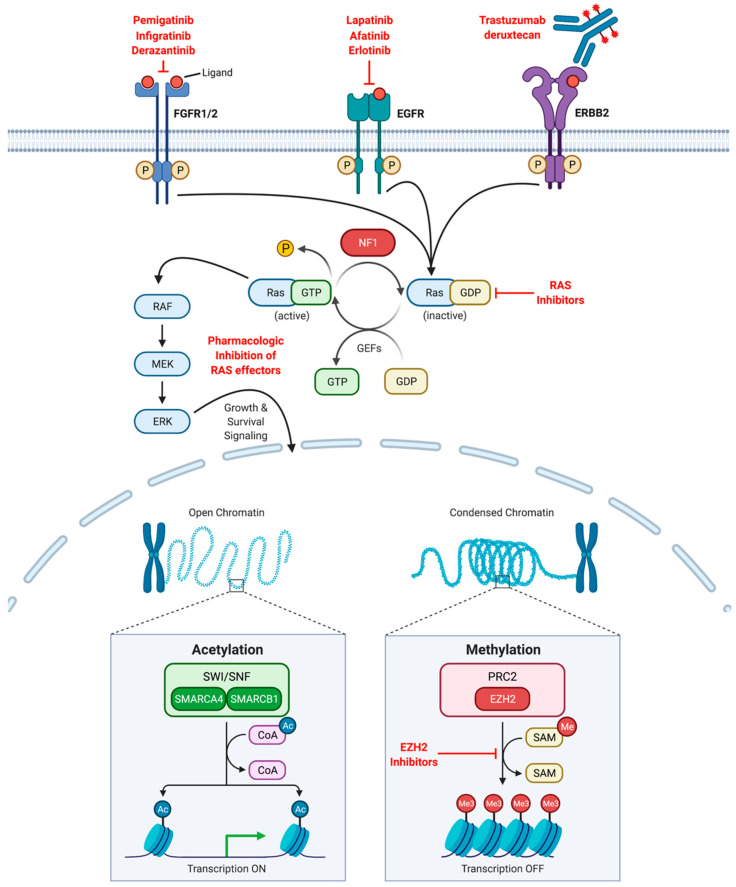
Potential therapeutic targets for ASCC. Cell surface receptors recurrently mutated in ASCC include fibroblast growth factor receptors 1 and 2 (FGFR1/2), epidermal growth factor receptor (EGFR) and Erb-B2 (ERBB2) receptor tyrosine kinase. These result in constitutive growth signaling. NRAS and KRAS are also recurrently mutated. RAS is activated through the exchange of guanosine diphosphate (GDP) for guanosine triphosphate (GTP) by guanine nucleotide exchange factors (GEF). RAS typically catalyzes GTP to release a phosphate (P) residue converting it to an inactivated state. Catalysis is facilitated by GTP activating proteins such as NF-1, which is frequently inactivated in ASCC. NRAS and KRAS mutations typically result in impaired GTP catalysis resulting in constitutive downstream growth signaling. Under normal conditions, gene expression is regulated by both switch/sucrose nonfermentable (SWI/SNF) complex and polycomb repressor complex 2 (PRC2). Subunits of SWI/SNF, SMARCA4 and SMARCB1 are recurrently mutated resulting impairing histone acetyltransferase activity of SWI/SNF. This results in unopposed activity of PRC2-mediated histone methyltransferase by subunit enhancer of zeste homoglog 2 (EZH2). Indicated in red are several pharmacologic inhibitors of these pathways either in clinical practice or currently under investigation. Ac, acetyl group; CoA, coenzyme A; Me, methyl group; SAM, *S*-Adenosyl methionine.

**Figure 3 cancers-13-02180-f003:**
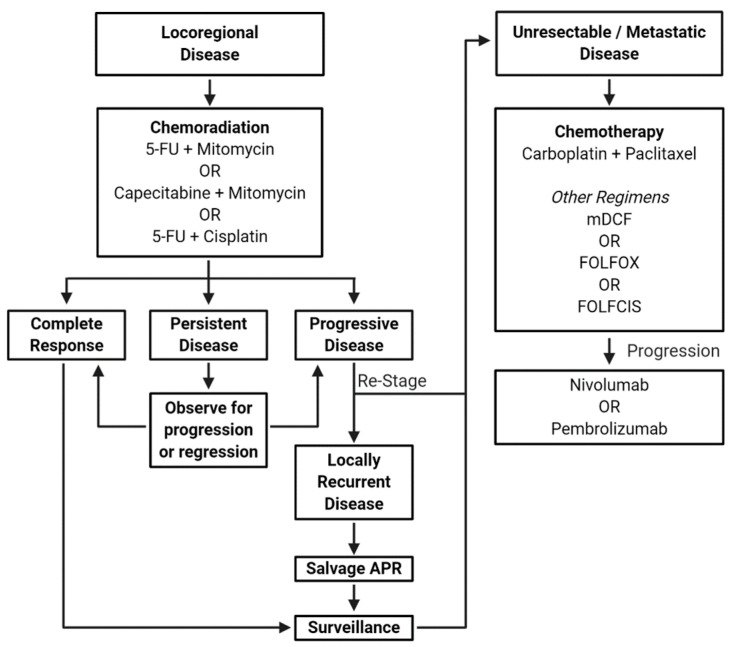
Current treatment paradigm for ASCC. 5-FU, 5-fluorouracili; APR, abdominal perineal resection; mDCF, modified docetaxel + cisplatin + 5-fluorouracil; FOLFOX, 5-fluorouracil + leucovorin + oxaliplatin; FOLFCIS, 5-fluorouracil + leucovorin + cisplatin.

**Table 1 cancers-13-02180-t001:** Landmark studies in management of locoregional ASCC.

Trial	N	Treatment Arms	Outcomes
EORTC 22861 [9]	110	Randomized phase III study comparing 5-FU + mitomycin with radiation vs. radiation alone	Improved CR rate (80% vs. 54%) Improved locoregional recurrence rate by 18% (*p* = 0.02) Improved colostomy-free interval by 32% (*p* = 0.002) Improved PFS (*p* = 0.05)
ACT I [11]	500	Randomized phase III study comparing 5-FU + mitomycin with radiation vs. radiation alone	Primary endpoint of local-failure rate at 3.5 years was reduced by 46% (HR 0.54, 95% CI: 0.42–0.69, *p* < 0.0001) Median follow-up of 13 years: Reduced in locoregional relapse by 25% (HR 0.46, 95% CI: 0.35–0.60) Reduced ASCC death by 12.5% (HR 0.67, 95% CI: 0.51–0.88) Improved median OS at 7.6 vs. 5.4 years (HR 0.86, 95% CI: 0.7–1.04)
RTOG 87-04/ECOG 1289 [12]	310	Randomized phase III study comparing chemoradiation with 5-FU + mitomycin vs. 5-FU alone	Improved colostomy-free survival (71% vs. 59%, *p* = 0.014) Improved DFS (73% vs. 51%, *p* = 0.0003)
EXTRA [19]	31	Single-arm phase II study using capecitabine + mitomycin chemoradiation	Complete response rate was 77% Approximately 10% locoregional relapses at median follow-up of 14 months
[20]	43	Single-arm phase II study using capecitabine-based chemoradiation	Primary endpoint of local control at six months was 86% (95% CI: 0.72–0.94)
ACT II [21]	940	Randomized phase III, 2 × 2 factorial design, comparing chemoradiation with mitomycin + 5-FU vs. cisplatin + 5-FU with or without maintenance chemo	Comparing mitomycin + 5-FU and cisplatin + 5-FU Primary endpoint of CR rates at 26 weeks was not significantly different (90.5 vs. 89.6%, 95% CI −4.9–3.1, *p* = 0.64) Comparing with or without maintenance chemotherapy: No significant difference in three-year PFS at 74% (95% CI: 69–77) and 73% (95% CI: 68–77) (HR 0.95, 95% CI: 0.75–1.21, *p* = 0.70)
[22]	19	Phase II pilot study treating with 5-FU + mitomycin + cisplatin chemoradiation	Sixteen (84%) developed grade 3/4 toxicities with one patient dying as a complication of treatment At median follow-up of 79 months, 84% remained disease-free Approximately 10% locoregional relapses at median follow-up of 14 months
RTOG 98-11 [23]	649	Randomized phase III study comparing chemoradiation with 5-FU and mitomycin vs. 5-FU and cisplatin	Primary endpoint of five-year DFS improved at 67.8% vs. 57.8% (*p* = 0.006) Improved five-year median OS of 78.3% vs. 70.7% (*p* = 0.026)
ACCORD 03 [24]	307	Randomized phase III study comparing chemoradiation with or without induction 5-FU and cisplatin	Primary endpoint of five-year colostomy-free survival was 76.5% (95% CI: 68.6–83.0) vs. 75% (95% CI: 67.0–81.5, *p* = 0.37)

5-FU, 5-fluorouracil; CI, confidence interval; CR, complete response; DFS, disease-free survival; HR, hazard ratio; N, number of patients; OS, overall survival; PFS, progression-free survival.

**Table 2 cancers-13-02180-t002:** Landmark Studies in Management of Metastatic ASCC.

Trial	N	Treatment Arms	Outcomes
Epitopes-HPV02 [32]	66	Nonrandomized, single-arm phase II treating with either DCF or mDCF with allocation determined by age and PS	Primary endpoint 12-month PFS was not significantly different (61% had progressed with DCF while 60% had progressed with mDCF) Improved locoregional recurrence rate by 18% (*p* = 0.02) Improved colostomy-free interval by 32% (*p* = 0.002) Improved PFS (*p* = 0.05)
InterAAct [33]	91	Randomized phase II study comparing carboplatin + paclitaxel vs. cisplatin + 5-FU	Comparable ORR at 59% (95% CI: 42.1–74.4%) vs. 57% (95% CI: 39.4–73.7%) Improved PFS (8.1 vs. 5.7 months) and OS (20 vs. 12.3 months) (HR 2.00, 95% CI: 1.15–3.47, *p* = 0.014) with carboplatin + paclitaxel Increased serious adverse events cisplatin + 5-FU arm (62% vs. 32%, *p* = 0.016)
KEYNOTE-028 [34]	25	Single-arm phase Ib study of pembrolizumab in second line	Primary endpoint of ORR was 17% (95% CI: 5–37%) Duration of response that was not reached at median follow-up of 10.6 months Median PFS was 3.0 months (95% CI: 1.7–7.3 months) Median OS was 9.3 months (95% CI: 5.9 months—not available)
NCI9673 [35]	37	Single-arm phase II study of nivolumab in second line	RR was 24% (95% CI: 15–33)

5-FU, 5-fluorouracil; CI, confidence interval; CR, complete response; DCF, docetaxel + cisplatin + 5-fluorouracil; mDCF, modified DCF; N, number of patients; ORR, overall response rate; OS, overall survival; PFS, progression-free survival; PS, performance status; RR, response rate.

**Table 3 cancers-13-02180-t003:** Studies testing EGFR targeted therapy in ASCC.

Trial	N	Treatment Arms	Outcomes
[46]	21	Single-arm phase I study with chemoradiation with 5-FU, cisplatin and cetuximab	RR of 95% (95% CI: 78–99%) At median follow-up of 43.4 months, three-year locoregional control was 64.2% (95% CI: 57.15–70.40%) Closed prematurely due to high rates of grade 3/4 adverse events
ACCORD 16 [47]	16	Single-arm phase II study with chemoradiation with 5-FU, cisplatin and cetuximab	One-year colostomy-free survival 67% (95% CI: 40–80%) PFS of 62% (95% CI: 36–82%) OS of 92% (95% CI: 67–99%) Prematurely closed due to frequent serious adverse events
E3205 [48]	61	Single-arm phase II study of pembrolizumab in second line	Primary endpoint of 3-year locoregional failure rate 23% (95% CI: 13–36%, *p* = 0.03) Three-year PFS and OS were 68% (95% CI: 55–79%) and 83% (95% CI: 71–91%), respectively Grade 4 toxicities occurred in 32% of patients with 5% treatment-related deaths
AMC045 [49]	37	Single-arm phase II study of nivolumab in second line	Three-year locoregional failure rate was 42% (95% CI: 28–56%, *p* = 0.9) Three-year PFS and OS were 72% (95% CI: 56–84%) and 79% (95% CI: 63–89%), respectively Grade 4 toxicities occurred in 26% of patients with 4% treatment-related deaths

5-FU, 5-fluorouracil; CI, confidence interval; N, number of patients; OS, overall survival; PFS, progression-free survival; RR, response rate.

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
