# Peer review of "Research on Anal Squamous Cell Carcinoma: Systemic Therapy Strategies for Anal Cancer"

_cancers, 2021, doi:10.3390/cancers13092180_

Round 1
Reviewer 1 Report
ok
Reviewer 2 Report
the authors appropriately addressed previous concerns and have added significantly more regarding other novel therapeutic targets in ASCC. They have also improved the review of past literature in multiple tables.
This manuscript is a resubmission of an earlier submission. The following is a list of the peer review reports and author responses from that submission.
Round 1
Reviewer 1 Report
- This is a review of systemic therapy in the management of anal cancers. The authors provide a thorough review of available data regarding the role of chemotherapy in all stages of anal cancer. While this is a thorough review, it is unclear if this information is novel enough to justify an additional publication. There are already reviews summarizing systemic therapies and new advancements in anal cancer, for example (PMID 29728940, 29494827) and many textbooks and uptodate already include summaries of standard of care anal cancer treatment, which makes up the first third of the paper. What about this publication is novel?
Minor comments
- The paragraphs are long and read like lists of studies, endpoints, and results. There should be some more description in the text of the main findings of these studies, with most of the studies and numerical results put into tables for more easy reading.
- overuse of the word ultimately in lines 82-91 - used 3 times in close proximity, try to find different wording
- one of 2 total figures shows the PI3K/Akt/MTOR signaling axis. Additionally a large section of the paper (Line 286-354) is also soley devoted to this one pathway, and most of the data is preclinical. There are multiple other novel systemic therapies being explored in anal cancer, why put so much focus on this one?
Author Response
We greatly appreciate the time you took to assess our manuscript and your thoughtful, constructive feedback. Our point-by-point responses are as follows:
- While this is a thorough review, it is unclear if this information is novel enough to justify an additional publication.
Response: We agree with your feedback regarding the overlap of this manuscript with others discussing management of ASCC. However, to provide further context, this manuscript is an invited review for a special issue of Cancers dedicated to “Research on Anal Squamous Cell Carcinoma”. Specifically, we were invited by the guest editor to write a review on the use of systemic therapy in ASCC management. While you are certainly correct that the state of systemic therapy for locoregional and metastatic disease has not dramatically changed since the publication of the reviews you cited, our review of immunotherapy, HPV-based vaccines and targeted therapy is more up to date.
- The paragraphs are long and read like lists of studies, endpoints, and results. There should be some more description in the text of the main findings of these studies, with most of the studies and numerical results put into tables for more easy reading.
Response: Thank you for this comment, it is an excellent suggestion. Much of the numerical results of the various cited studies have been moved to tables 1-3 as suggested. Table 1 summarizes results/endpoints of studies investigating treatment for locoregional disease while Table 2 focuses on metastatic disease. Table 3 summarizes trials involving EGFR blockade.
- Overuse of the word ultimately in lines 82-91 – used 3 times in close proximity, try to find different wording.
Response: We appreciate you bringing this to our attention. The wording has been altered.
- One of 2 total figures shows the PI3K/Akt/MTOR signaling axis. Additionally a large section of the paper (Line 286-354) is also solely devoted to this one pathway, and most of the data is pre-clinical. There are multiple other novel systemic therapies being explored in anal cancer, why put so much focus on this one?
Response: This is an excellent point. While the “Precision Medicine and Targeted Therapy” section of the manuscript highlights the shortcomings of EGFR inhibitors, the remainder largely focuses on the PI3K/Akt signaling axis. The rationale for this is highlighted in the third paragraph. Across the various sequencing studies of ASCC the majority of identified somatic variants are predicted to enhance PI3K/Akt/mTOR signaling. In addition, I defend the inclusion of pre-clinical data as it serves as proof-of-concept for rationally developed novel therapeutic strategies and better informs biological understanding of disease. PI3K/Akt/mTOR signaling axis abrogation as a novel, targeted therapeutic strategy for ASCC is also the most studied in preclinical models.
Nonetheless, we do agree that additional novel systemic therapies being explored in ASCC should be highlighted. A discussion of these potential targets is now included in the targeted therapy section of the manuscript. A graphical depiction of these strategies has also been added.
Reviewer 2 Report
The authors provided a comprehensive review on the therapies for a rare malignancy, anal squamous cell carcinoma (ASCC), including chemotherapy, immunotherapy, and targeted therapy. The authors summarized the current treatment paradigm and also provided perspectives on new treatment strategies. The manuscript is well organized and easy to follow.
Author Response
We greatly appreciate your consideration of our review.
Reviewer 3 Report
In the review article titled “Research on Anal squamous Cell Carcinoma: Systemic Therapy strategies for Anal Cancer”. In this article, the Authors discussed anal squamous Cell Carcinoma (ASCC) which is a rare malignancy with most cases associated with Human papillomavirus and increased incidence in immunocompromised patients. They thoroughly discussed the shortcomings and the use of systemic therapy in the management of ASCC. The authors emphasized the evolving role of novel treatment strategies such as immune checkpoint inhibitors, HPV-based vaccine, and molecularly targeted therapies with rigorous literature support.
Overall, the study is interesting and comprehensive and I am sure it will open new vistas in the field of ASCC management. The article is well written and easy to understand all the aspects discussed.
Therefore, for these reasons, I would suggest the editor to consider this article for publication in cancers without revision.
Author Response
We sincerely thank you for taking the time to consider our review.
Reviewer 4 Report
The review is well organized and covers widely different topics of an uncommon disease with a growing incidence. Although many aspects regarding systemic therapy are very well treated, the core of the standard treatment is chemoradiation. It could be also useful adding some evidences ( a small paragraph) about radiotherapy.
Author Response
We greatly appreciate your consideration of our review. We completely agree with your feedback regarding the importance of radiation in the chemoradiation regimens. However, to provide further context, this manuscript was an invited review for a special issue of Cancers dedicated to “Research on Anal Squamous Cell Carcinoma”. Therefore, the role of radiation therapy will be more thoroughly discussed in other articles of the issue. This is the rationale for omitting the topic of radiation in this manuscript. Once again, thank you for taking the time to assess our review.
Reviewer 5 Report
This is a well written review of the anal squamous cell cancer in a very systematic view. The article flows well with role of chemoradiation, role of induction or maintenance chemotherapy, options for metastatic disease, role of immunotherapy as well as precision medicine and targeted therapy.
Line 140 mentioning figure 1 in the systemic therapy for metastatic disease should be figure 2 which is the flow chart describing the treatment paradigm.
I did not find any issues with the references.
Author Response
Thank you for taking the time to assess our review. The mistake noted in Line 140 of the manuscript has now been corrected.